# Studies of salt and stress sensitivity on arterial pressure in renin-b deficient mice

**Pablo Nakagawa**[ID]*, **Javier Gomez, Ko-Ting Lu, Justin L. Grobe, Curt D. Sigmund**[ID]

Department of Physiology, Medical College of Wisconsin, Milwaukee, Wisconsin, United States of America

* pnakagawa@mcw.edu

**Data Availability Statement:** All relevant data are within the manuscript.

**Funding:** This work was supported through research grants from the National Institutes of Health (NIH) to CDS (HL084207 and HL144807),

## Abstract

Excessive sodium intake is known to increase the risk for hypertension, heart disease, and stroke. Individuals who are more susceptible to the effects of high salt are at higher risk for cardiovascular diseases even independent of their blood pressure status. Local activation of the renin-angiotensin system (RAS) in the brain, among other mechanisms, has been hypothesized to play a key role in contributing to salt balance. We have previously shown that deletion of the alternative renin isoform termed renin-b disinhibits the classical renin-a encoding preprorenin in the brain resulting in elevated brain RAS activity. Thus, we hypothesized that renin-b deficiency results in higher susceptibility to salt-induced elevation in blood pressure. Telemetry implanted Ren-b$^{Null}$ and wildtype littermate mice were first offered a low salt diet for a week and subsequently a high salt diet for another week. A high salt diet induced a mild blood pressure elevation in both Ren-b$^{Null}$ and wildtype mice, but mice lacking renin-b did not exhibit an exaggerated pressor response. When renin-b deficient mice were exposed to a high salt diet for a longer duration (4 weeks), there was a trend for increased myocardial enlargement in Ren-b$^{Null}$ mice when compared with control mice, but this did not reach statistical significance. Multiple studies have also demonstrated the association of environmental stress with hypertension. Activation of the RAS in the rostral ventro-lateral medulla and the hypothalamus is required for stress-induced hypertension. Thus, we next questioned whether the lack of renin-b would result in exacerbated response to an acute restraint-stress. Wildtype and Ren-b$^{Null}$ mice equally exhibited elevated blood pressure in response to restraint-stress, which was similar in mice fed either a low or high salt diet. These studies suggest that mechanisms unrelated to salt and acute stress alter the cardiovascular phenotype in mice lacking renin-b.

## Introduction

High blood pressure is a leading cause of complications from heart disease, stroke, and kidney disease. Importantly, it has been shown that in both normotensive and hypertensive subjects, increased sensitivity to sodium increases the risk for cardiovascular mortality and morbidity [1]. The increased prevalence of hypertension in the last century has been attributed in part to the higher consumption of dietary salt [2]. Moreover, there is strong evidence in animal

JLG (HL134850), PN (HL153101) and grants from the American Heart Association to CDS (15SFRN23480000), JLGrobe (18EIA33890055). PN was funded with training awards from the NIH for this study (2T32HL007121041, 4T32DK007690, and 5T32HL134643).

**Competing interests:** We declare that Curt D. Sigmund was a member of the Scientific Advisory Board for Ionis Pharmaceuticals. His contributions to that board are unrelated to the content of this article and this does not alter our adherence to PLOS ONE policies on sharing data and materials. The other authors have no conflicts.

models and human for a causative role of dietary sodium intake in the development of hypertension [2–4]. Dietary sodium reduction in combination with the DASH (dietary approaches to stop hypertension) diet is a powerful approach to control blood pressure in hypertensive patients [5,6].

According to Guyton and Coleman's hypothesis, salt-sensitive hypertension can develop when there is an impaired excretory ability of the kidney that shifts the relation between sodium excretion and arterial pressure toward higher values [7]. Recently, Fujita et al. identified a key mechanism implicating renal β2-sympathetic stimulation contributing to the development of salt-sensitive hypertension [8]. Numerous studies have identified neurogenic actions of elevated salt intake [9,10]. For instance, elevated dietary sodium intake sensitizes sympathetic neurons in response to injection of L-glutamate into the rostral ventrolateral medulla [11]. In addition to the effects on arterial pressure, high salt diet can also adversely affect several organs including the vasculature, heart, and kidneys. Interestingly, some of these effects appear to be independent of blood pressure status [12].

The renin-angiotensin system (RAS) is recognized as a powerful circulating hormone system regulating blood pressure and sodium homeostasis. Although high salt intake suppresses the circulating classical RAS, the deterioration of renal function in sodium loaded animals can be attenuated by the blockade of the RAS [13]. Thus, it has been proposed that activation of a locally activated RAS within tissues, which acts independently of the circulating RAS, can be of major importance in the regulation of the angiotensin (Ang) II generation in the kidney, brain, vasculature, and heart [14–18]. Renin is the rate-limiting step of Ang II synthesis and exists in at least two isoforms. The classical renin isoform encoding preprorenin termed "renin-a" is primarily expressed by the juxtaglomerular cells of the kidney and is released into the circulatory system in response to a number of stimuli. On the contrary, the novel alternative renin isoform, termed "renin-b", is mainly expressed in the brain, heart, and adrenal gland [19–21]. Current evidence suggests that renin-b is unlikely to be secreted as it lacks a secretory peptide. Our previous data suggest that renin-b maintains brain RAS activity at normal physiological levels by suppressing the classical renin isoform and that the genetic removal of renin-b triggers a mild increase in blood pressure and sympathetic nerve activity [22–24]. The sustained suppression of the brain RAS by renin-b in healthy individuals may have important implications in maintaining the cardiovascular homeostasis during different hypertension-triggering stimuli such as high salt diet and other environmental stressors. Blood pressure elevation induced by the exposure to physical stress can be ameliorated when the brain RAS is suppressed [25,26]. Given the rationale that hypertension-prone animal models exhibit a greater increase in blood pressure and heart rate in response to restraint-stress and air-jet stress [27,28], we hypothesized that mice lacking renin-b would be more responsive to an acute environmental stressor as consequence of brain RAS disinhibition.

## Methods

### Animals

All experiments were conducted using mice lacking exon 1b of the renin gene on the C57BL/6 genetic background as previously described [22]. Experimental animals were generated by intercrossing heterozygotes to generate homozygous control wildtypes (WT) and homozygous Ren-b^Null mice. Only males were used except where indicated. Males were preferred in this study because we previously observed that Ren-b^Null males exhibit an exacerbated phenotype compared to females [23]. A room with low levels of noise and traffic was designated for these experiments. Animals received standard laboratory chow (NIH-31 Modified Open Formula Mouse/Rat sterilizable diet; Teklad catalog # 7913; Harlan Teklad) until exposure to low and

high salt diets. All experiments were conducted with the approval of the University of Iowa and Medical College of Wisconsin Animal Care and Use Committees and were performed in accordance with the National Institutes of Health Guide for the Care and Use of Laboratory Animals.

## Blood pressure measurements

Twenty-four-hour systolic, diastolic, and mean blood pressures and heart rate telemetric data were acquired by radiotelemetry. Telemetry implantation surgeries and telemetric data acquisition were performed by a surgeon who was blinded to genotype. Anesthesia was induced with ketamine (90 mg/kg i.p.)/xylazine (5 mg/kg i.p.) to implant the telemetry catheters. Then, a 1–2 cm incision was made to expose the internal carotid artery. A TA11PA-C10 radiotelemetry catheter (DSI, New Brighton, MN) was inserted in the left carotid artery and advanced into the transverse aorta. The catheter was secured with three sutures and the incision was sutured with 6–0 surgical silk. A triple antibiotic ointment, Vetropolycin (Dechra Pharmaceuticals PLC, Northwich, United Kingdom), was applied to the wound. Flunixin (2.5 mg/kg, i.p.) was immediately applied after the surgery and again 24 hours later. Then, the animals were housed in a cage placed on a heating pad overnight with easy access to food and water. Once awake from the insertion of the radiotelemetry implant procedure, the animals were housed individually in a regular cage placed on top of the telemetry receiver plate with free access to water and a standard chow diet containing 0.3% Sodium. Baseline blood pressure and heart rate data were acquired for 3 days after 10 days of post-surgical recovery. Blood pressure and heart rate data were recorded for 10 seconds every 5 minutes for 24 hours and averaged daily, hourly, and during the light (5 AM to 5 PM) and dark phase (5 PM to 5 AM).

## Assessment of salt sensitivity

**Experiment 1.** Telemetry implanted mice underwent 3-day baseline blood pressure and heart rate data acquisition under a normal salt diet containing 0.8% NaCl (NIH-31 Modified Open Formula Mouse/Rat sterilizable diet; Harlan Teklad). The standard diet containing normal salt was then replaced with a salt-deficient chow diet containing 0.01–0.02% sodium (TD08290, Envigo, Indianapolis, IN). Data were acquired for 7 days on a low salt diet. Then, the diet was switched to a high salt chow containing 4% NaCl (TD03095, Envigo, Indianapolis, IN) and data were acquired for 7 additional days (Fig 1).

**Experiment 2.** Renal concentrating response to acute hypo and hyperosmotic load was assessed in a cohort of male and female WT, heterozygous, and Ren-b$^{Null}$ mice. Mice were forced to empty the bladder by immobilizing them. Then, hypotonic saline (500 microliters of 0.03 mol/L NaCl; intraperitoneal) was applied and the animal was placed in a 1 L beaker for 60 minutes. Subsequently, the same procedure was performed for a hypertonic saline challenge (500 microliters of 0.3 mol/L NaCl; intraperitoneal). Urine was collected from the beaker and the osmolality was assessed using a freezing point depression multi-sample osmometer (OsmoPro, Advanced Instruments, Norwood, MA). These experiments were performed from 9 AM to 12 PM.

**Experiment 3.** A separate cohort of mice without telemeters were placed in metabolic cages for urine collection at baseline, under a standard chow diet (0.8% NaCl), and after 4 weeks on a high salt diet (4% NaCl). Food and water intake, urinary volume, and protein excretion were assessed. Protein concentration was measured using a commercially available protein determination kit from Thermo Scientific (catalog # 23225), which is based on the bicinchoninic acid method. By the end of week 4, mice were anesthetized with pentobarbital and blood samples were obtained from vena cava using a heparinized syringe for measurement

# Experimental design

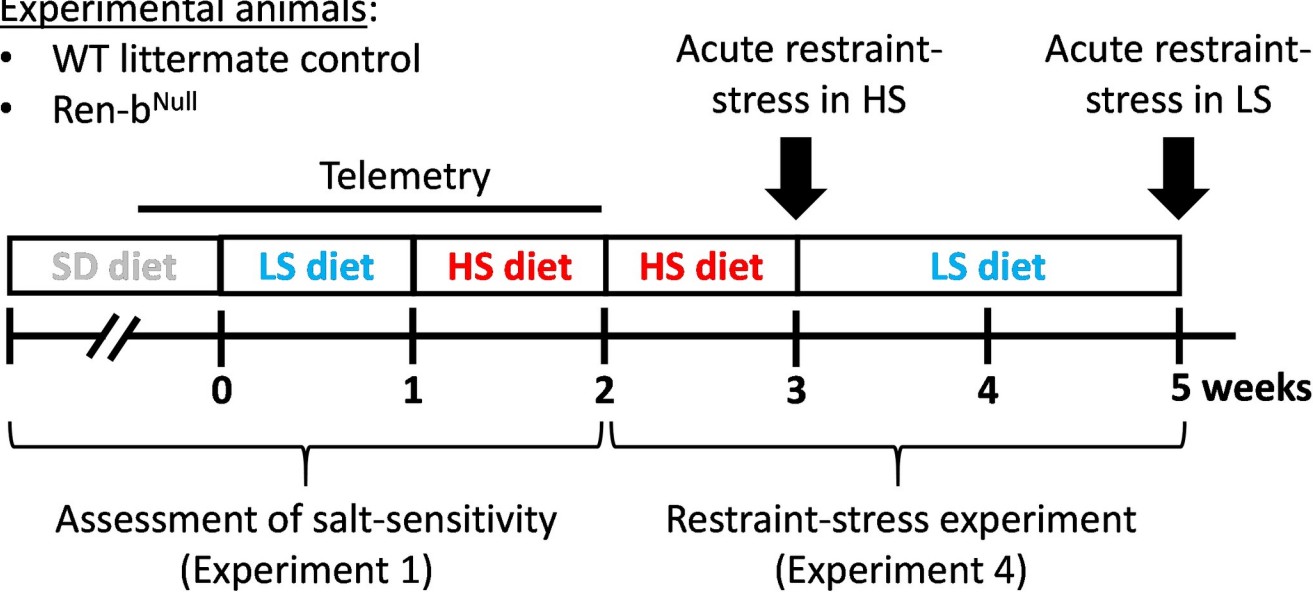

**Fig 1. Experimental protocol.** The assessment of salt sensitivity (Experiment 1) and susceptibility to restraint-stress (Experiment 4) were performed in the same animals. Male Ren-b$^{Null}$ or wildtype (WT) littermate controls were instrumented with a radiotelemetry catheter and housed in a quiet environment for blood pressure and heart rate measurements. In experiment 1, baseline arterial blood pressure and heart rate were recorded for 3 days under a standard salt diet (SD, 0.8% NaCl). Subsequently, the diet was switched to a chow containing low sodium (LS, 0.02% NaCl), and data were acquired for 7 additional days. Finally, the diet was switched from low to high salt containing diet (HSD, 4% NaCl), and data were acquired for 7 additional days. To study responses to acute restraint stress in both HSD and LSD, the animals were continued on HSD for an additional week. By the end of the second week on HSD, mice were placed to an acrylic restrainer to assess the acute cardiovascular responses to restraint-stress. Next, mice were fed LSD for two weeks and the acute responses to restraint-stress were reassessed again.

of plasma renin. Subsequently, mice were sacrificed, and kidneys, heart, liver, spleen, and lungs were collected and weighed.

### Responses to restraint stress in mice fed high or low salt diets

**Experiment 4.** Telemetry implanted mice tested for salt-sensitivity in experiment 1 were subsequently subjected to acute experiments to assess sensitization to restraint-stress (Fig 1). First, we evaluated the cardiovascular responses to an acute immobilization protocol. WT and Ren-b$^{Null}$ mice fed high salt diet were placed for 10 minutes in a restrainer, while telemetric blood pressure and heart rates were monitored. Then, the diet was switched back to a low salt diet and the same procedure was performed 14 days later. Baseline telemetric blood pressure and heart rate were acquired for 10 minutes before mice were placed in the restrainer. Once in the restrainer, data were acquired for 10 additional minutes. A separate cohort of males and females fed with a normal salt diet was used to study the metabolic responses to restraint-stress. Blood glucose was measured at baseline before restraint-stress and induced a restrain-stress for 30 minutes. Finally, mice were released to the home cage and blood glucose was measured again during recovery. Every 15 minutes blood samples were obtained from tail veins in conscious mice and blood glucose levels were determined using a glucometer (One Touch Ultra 1, LifeScan, Inc.). Mice were sacrificed by decapitation at 30 minutes post-recovery phase and blood was collected for renin and glucocorticoid determinations.

## Plasma renin measurements

Heparinized blood was collected in 1.5 ml tubes from the inferior vena cava in anesthetized mice or trunk after decapitation. Blood samples were centrifuged at 4000 rpm for 2 min at 4˚C to isolate plasma. Plasma renin was measured using a commercially available Sandwich-based ELISA kit (Raybiotech, Peachtree Corners, GA) following the manufacturer's recommended instructions. Samples were processed and analyzed by a blinded operator.

## Expression of renin in the central nervous system and adrenal glands

**Experiment 5.** Wildtype and Ren-b$^{Null}$ mice fed a standard chow diet (0.8% NaCl) were sacrificed by injecting pentobarbital overdose (150 mg/kg). The animals were decapitated, the brains were excised from the skull and the adrenal glands collected and frozen in liquid nitrogen. The hypothalamus was carefully dissected and immediately immersed in liquid nitrogen. Some additional brains were embedded in Tissue-Tek OCT compound (Fisher Scientific, Waltham, MA) for subsequent rostral ventrolateral medulla (RVLM) sample collection using the puncture technique. With the help of a polypropylene pellet pestle the whole hypothalamic tissue, the RVLM punches, or adrenal glands were homogenized in TRIzol reagent (Thermo Fisher Scientific, Waltham, NA). The total mRNA was isolated with a Purelink RNA Mini kit (Invitrogen, Carlsbad, CA) and eluted in 30 microliters of RNase free water. Next, five hundred nanograms of total RNA was reverse transcribed using an iScript kit (Biorad, Hercules, CA). In the hypothalamic and RVLM samples, a real-time PCR was performed utilizing Taq-Man probes and Fast Advanced Master mix (catalog # 4444557) from Applied Biosciences following the manufacturer's protocol. The catalog numbers of Taqman probes were Ren1: Mm02342887_mH and Gapdh: Mm99999915_g1. In adrenal gland samples, a real-time PCR was performed using Applied Biosystems SYBR Green master mix and the following primer sets:

## Ren1

Forward: 5'-TCCCGGACAGAAGGCATTTTC-3'

Reverse: 5'-TGTGTCACAGTGATTCCACCCACA-3'

## Renin-a

Forward: 5'-GCACCTTCAGTCTCCCAACAC-3'

Reverse: 5'-TCCCGGACAGAAGGCATTTTC-3'

## 18s

Forward: 5'-CGCTTCCTTACCTGGTTGAT-3'

Reverse: 5'-GAGCGACCAAAGGAACCATA-3'

The data analysis was performed using the Livak method [29].

## Statistical analysis

All data are presented as mean±SEM. Data were analyzed using GraphPad Prism software. All telemetric data, parameters derived from metabolic cage studies and osmotic challenge tests,

plasma glucose, renin and corticosterone measurements were analyzed using two-way ANOVA with repeated measures, followed by Sidak's multiple comparison procedures. Tissue weights were analyzed using unpaired t-test. A *P value* of less than 0.05 was considered significant. Individual data points were plotted in dot/whisker plots if possible.

## Results

### Retrospective analysis of the arterial blood pressure in Ren-b<sup>Null</sup> cohorts

Retrospective analysis of blood pressure in five cohorts of Ren-b[Null] and WT mice (listed in order of experimentation) indicated that mice lacking Ren-b exhibit a high degree of variability in blood pressure (range = 27.7 mmHg, Fig 2). The range of blood pressure in control mice was much smaller (5.5 mmHg). We hypothesized that these differences might be attributed to environmental factors as a review of conditions under which the cohorts were studied revealed that cohorts 1, 2, 3, and 4 were housed in a large room with an elevated level of noise and traffic (data collected between March 2014 and October 2015 [22]), while mice in cohort 5 were housed in a smaller and newer quiet animal facility located in a different building (data collected from 2/2017 to 11/2017 [30]). Cohorts 2 and 4 were exposed to an additional stress as they were subjected to a surgical implantation of a stainless-steel intracerebroventricular

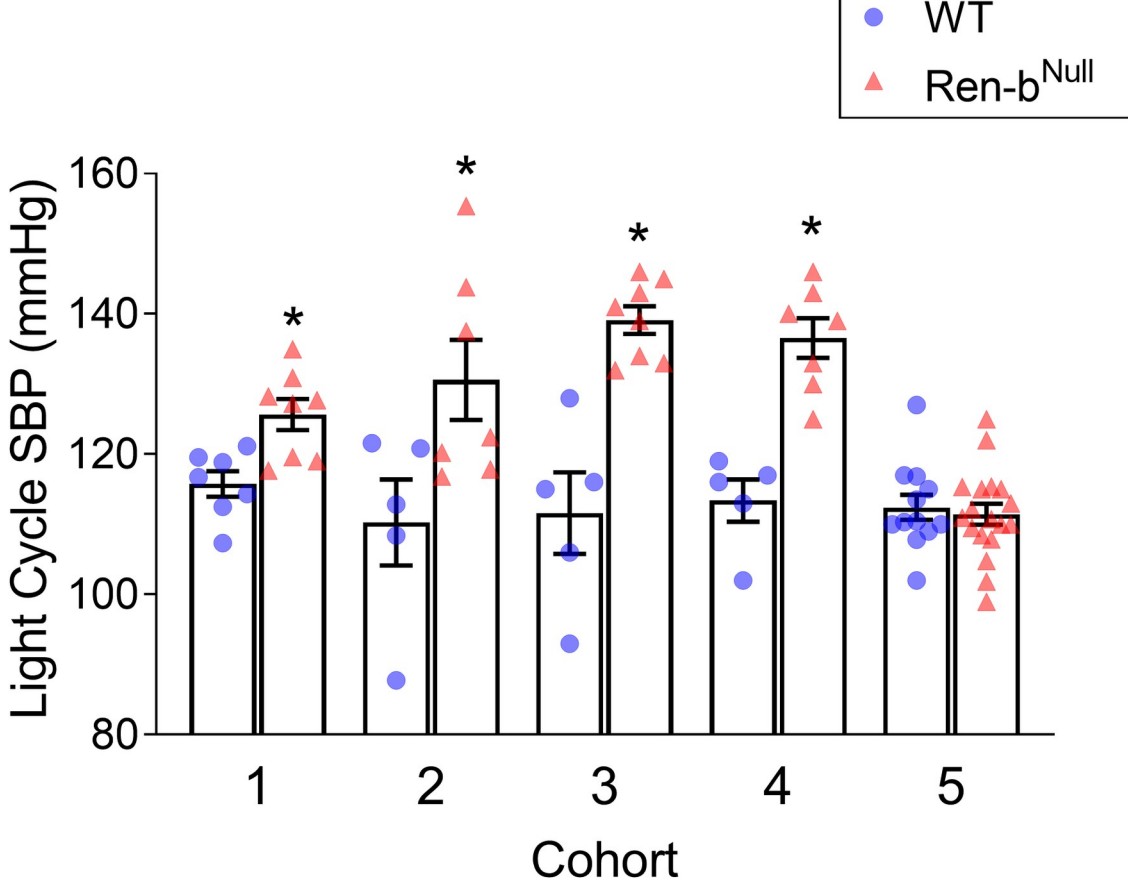

**Fig 2. Cohort analysis of arterial blood pressure.** Reanalysis and comparison of the telemetric blood pressure in 5 independent cohorts of Ren-b[Null] and wildtype (WT) mice. Summary of the systolic blood pressure (SBP) during the light cycle is shown. This data was previously reported in references [22] and [30]. *P<0.05 vs WT by unpaired t-test.

cannula for subsequent acute administration of drugs. These data were previously reported [22,30]. Cohorts 3 and 4 were subjected to extensive animal handling during experiments. Noise, post-surgical stress and animal handling appeared associated with an exacerbated blood pressure increase in Ren-b^Null mice. Therefore, the aim of this study was to elucidate whether the manifestation of cardiovascular and renal phenotypes in Ren-b^Null mice are due to increased sensitivity of these animals to external stressors such as environmental stimuli (i.e., noise, diet, and other psychological stresses such as restraint or handling).

### Effect of high salt diet on blood pressure in Ren-b^Null mice

Wildtype and Ren-b^Null mice fed a normal salt diet were instrumented with telemetric implants and allowed to recover for 10 days in a quiet environment. As shown in Fig 3, baseline blood pressure was first measured for 3 days under a normal salt diet containing 0.3% sodium. Then, the diet was replaced from normal to a low salt diet containing chow (Na 0.01–0.02%) and blood pressure recordings continued for 7 additional days. Finally, mice were fed high-salt diet chow (NaCl 4%) for the last 7 days. The low salt diet had no effects on systolic blood pressure, diastolic blood pressure, mean blood pressure or heart rate compared with the baseline diet (Fig 3). Blood pressure was increased in both wildtype and Ren-b^Null mice in response to the high salt diet when compared with the baseline diet ($P<0.05$) or low salt diet ($P<0.05$). Heart rate was not altered. Contrary to our hypothesis, there was no difference in blood pressure or heart rate comparing wildtype and Ren-b^Null mice in any diet.

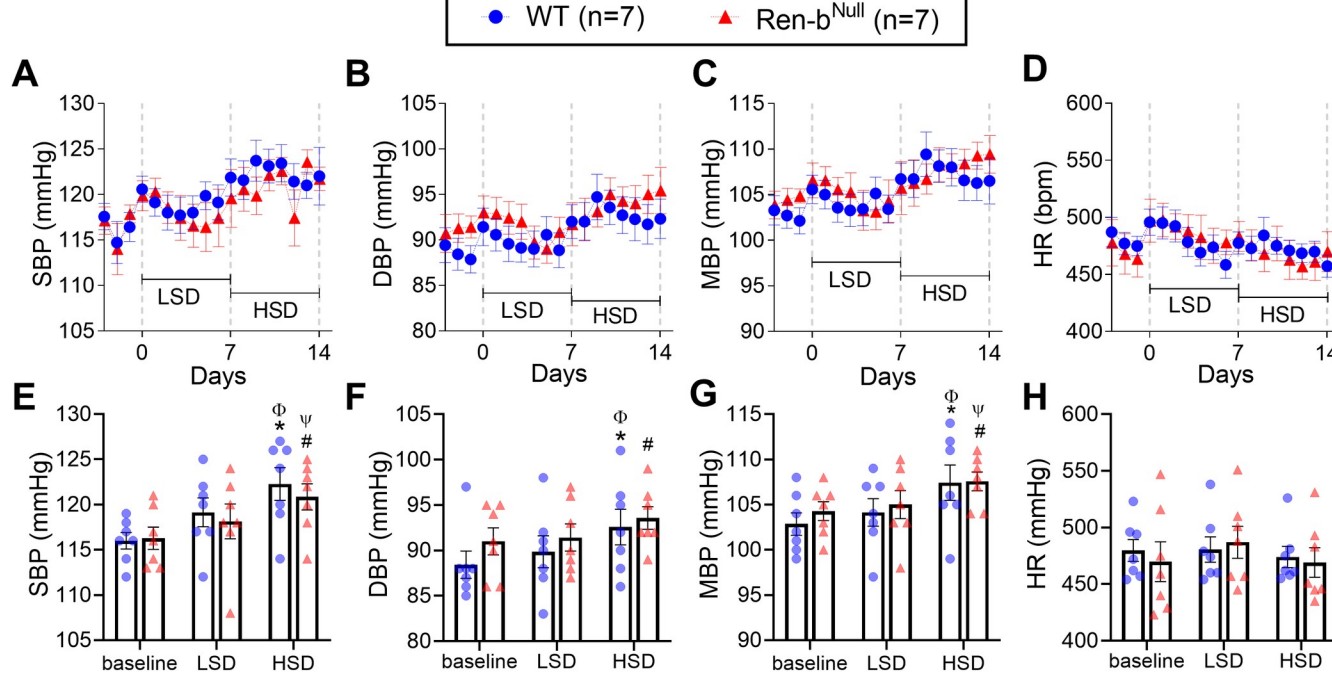

**Fig 3. Blood pressure response to salt.** Effect of salt intake on systolic blood pressure (SBP), mean blood pressure (MBP), diastolic blood pressure (DBP), and heart rate (HR) in male Ren-b^Null mice and wildtype littermate controls (WT) housed in a quiet environment. Following a 3-day baseline radiotelemetric data collection period, the diet was switched from a standard diet to a chow containing 0.02% NaCl (TD08290, LSD). Data were acquired for 7 days. Subsequently, mice were challenged with a diet containing 4% NaCl (TD03095, HSD), and data were collected for 7 additional days. Twenty-four-hour averaged A) SBP, B) DBP, C) MBP, and D) HR were expressed as mean ± standard error of the mean (top panel). For purposes of clarity and transparency, the individual points for weekly averaged E) SBP, F) DBP, G) MBP, and H) HR data were also plotted (bottom panel). Data were analyzed by two-way ANOVA with Sidak's multiple comparisons procedure. Adjusted p<0.05 was considered significant. *p<0.05 vs WT at baseline, #p<0.05 vs Ren-b^Null at baseline, Φp<0.05 vs WT fed LSD, ψp<0.05 vs Ren-b^Null fed LSD.

To evaluate the capacity of Ren-b$^{Null}$ mice to excrete a sodium load, mice were injected with either intraperitoneal hypo- or hyperosmotic NaCl solutions (0.5 ml of 0.03 and 0.3 mol/L NaCl, respectively). As expected, hyper-osmotic saline increased urinary osmolality compared with hypoosmotic saline injection. However, there was no significant difference in the urinary osmolality between wildtype, heterozygous, and Ren-b$^{Null}$ mice (Fig 4).

A separate cohort of Ren-b$^{Null}$ was placed in metabolic cages before and after 4 weeks on a high salt diet. One animal was excluded due to malocclusion. When compared with parameters at baseline, both WT and Ren-b$^{Null}$ mice fed a high salt diet for 4 weeks showed an increase in body weight and urine volume and a decrease in food intake and feces excretion (Table 1). Interestingly, Ren-b$^{Null}$ mice exhibited higher urinary protein excretion compared with wildtype mice at baseline. The level of protein excretion was maintained in Ren-b$^{Null}$ mice after high salt diet. On the contrary, high salt diet increased urinary protein excretion in WT mice. Since these mice were not implanted with telemeters it is unclear if in this cohort, the urinary protein levels were related to changes in blood pressure. In our previous studies we observed that Ren-b$^{Null}$ mice fed a normal salt diet exhibit a decreased in plasma renin [30]. However, in Ren-b$^{Null}$ mice fed a high salt diet we did not observe a statistical difference in renin levels among groups, although a trend toward decrease is evident (p = 0.094). Similarly, we observed a trend toward an increase in heart weight when comparing Ren-b$^{Null}$ with WT mice on a high salt diet (p = 0.08) (Table 2). No differences in body weight (WT = 28.36±1.16 g vs Ren-b$^{Null}$ = 29.84±1.45 g), kidney weight, liver weight, spleen weight, and lung weight were found among groups.

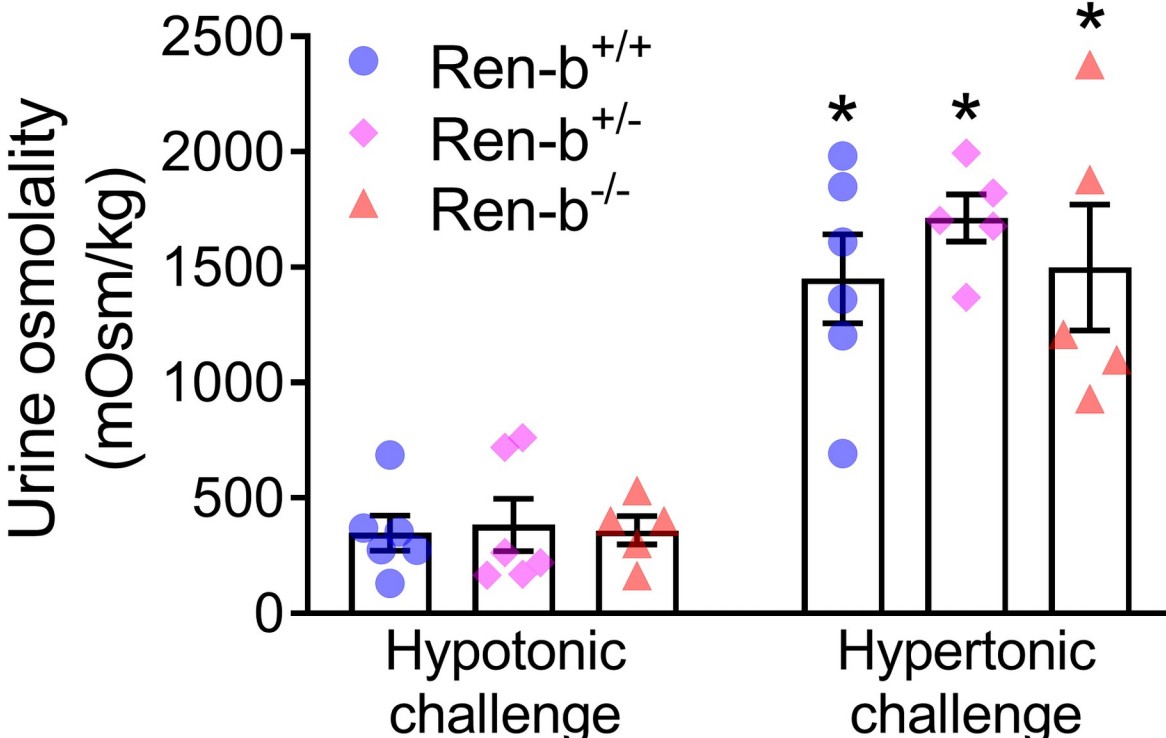

**Fig 4. Renal function.** Urine concentrating capacity was assessed in male and female wildtype (Ren-b$^{+/+}$), heterozygous Ren-b knockout (Ren-b$^{+/-}$) and homozygous Ren-b knockout mice (Ren-b$^{-/-}$). Urine was first collected for 60 minutes post-hypotonic saline challenge (500 microliters of 0.03 mol/L NaCl; intraperitoneal). Subsequently, the same procedure was performed for a hypertonic saline challenge (500 microliters of 0.3 mol/L NaCl; intraperitoneal). $^*$p<0.05 vs same genetic group on hypotonic challenge.

**Table 1. Male Wildtype (WT) and Ren-b<sup>Null</sup> mice were housed in metabolic cages to measure food intake, feces weight, water intake, urine volume, and urinary protein excretion at baseline and after 4 weeks on high salt diet (HSD, 4% NaCl).**

| | Baseline | | HSD | | p-value | | |
|---|---|---|---|---|---|---|---|
| | WT (n = 5) | Ren-b<sup>Null</sup> (n = 5) | WT (n = 5–6) | Ren-b<sup>Null</sup> (n = 5–6) | Timepoint | Genotype | Interaction |
| Body weight (g) | 27.26±0.60 | 26.54±0.75 | 28.10±0.48 | 27.60±0.84 | 0.002 | 0.533 | 0.620 |
| Food intake (g/day) | 2.63±0.23 | 2.90±0.33 | 2.16±0.19 | 2.10±0.21* | 0.008 | 0.744 | 0.383 |
| Water intake (ml/day) | 2.99±0.30 | 3.05±0.34 | 3.86±0.39 | 3.24±0.38 | 0.109 | 0.511 | 0.280 |
| Feces weight (g/day) | 0.70±0.09 | 0.85±0.12 | 0.44±0.03* | 0.49±0.07* | 0.001 | 0.374 | 0.488 |
| Urine volume (ml/day) | 0.65±0.16 | 0.61±0.07 | 1.13±0.12 | 1.39±0.34 | 0.015 | 0.604 | 0.492 |
| Urinary protein excretion (mg/day) | 11.74±5.62 | 51.66±9.70<sup>#</sup> | 31.61±7.37 | 44.12±4.52 | 0.374 | 0.009 | 0.070 |
| Plasma renin (ng/ml) | | | 44.8±12.8 | 24.3±7.0 | | 0.094 | |

At the end of the protocol renin was measured from plasma samples. Except plasma renin the data were analyzed by two-way ANOVA with Sidak's multiple comparisons procedure. Plasma renin data were analyzed by unpaired t-test.

*$p < 0.05$ vs same genotype at baseline and

<sup>#</sup>$p < 0.05$ vs WT at same timepoint.

### Responses to acute restraint-stress

Since there was no difference in blood pressure in response to the high salt diet, the same animals were further challenged with an acute restraint-stress. After 10 minutes of baseline recording, mice were quickly placed in an acrylic restrainer, and blood pressure and heart rate were recorded for additional 10 minutes. Restraint-stress induced a ~25 mmHg blood pressure elevation in both WT and Ren-b<sup>Null</sup> mice (Fig 5A–5C and 5E), which was associated with a robust tachycardic response (Fig 5D and 5F). No difference in the pressor responses to restraint-stress was observed in WT compared with Ren-b<sup>Null</sup> mice. Interestingly, Ren-b<sup>Null</sup> exhibited elevated heart rate ($P_{genotype} < 0.05$), which was more prominent at baseline as it achieved a similar increase compared with WT when exposed to restraint-stress. The individual values of blood pressure and heart rate before and during restraint-stress were plotted separately (Fig 5E and 5F). After the acute stress challenge was completed, the mice were shifted back to low salt diet for 2 weeks. Similarly, restraint-stress induced an increase in both blood pressure and heart rate in both groups, but no difference between WT and Ren-b<sup>Null</sup> was observed (Fig 6).

Previous cohorts of Ren-b<sup>Null</sup> mice exhibited resistance to body weight gain in response to high fat diet, which was attributed to elevated resting metabolic rate and sympathetic outflow to the brown adipose tissue [23]. Since restraint-stress induces hyperglycemia through the central catecholaminergic neuronal system located in RVLM [31], we evaluated in a separate

**Table 2. Body weight (left column) and tissue weights normalized by body weight (right column) in male Ren-b<sup>Null</sup> mice and wildtype littermate controls (WT) fed a high salt diet (HSD, 4% NaCl) for 4 weeks.**

| | Tissue weight (g) | | Tissue weight/body weight (mg/g) | |
|---|---|---|---|---|
| | WT+HSD (n = 6) | Ren-b<sup>Null</sup> +HSD (n = 5) | WT+HSD (n = 6) | Ren-b<sup>Null</sup> +HSD (n = 5) |
| Heart | 0.125±0.003 | 0.130±0.003 | 4.26±0.25 | 4.72±0.10 |
| Lungs | 0.157±0.006 | 0.150±0.010 | 5.34±0.17 | 5.43±0.30 |
| Kidneys | 0.177±0.009 | 0.173±0.006 | 6.01±0.21 | 6.27±0.06 |
| Liver | 1.217±0.084 | 1.256±0.066 | 41.48±2.96 | 45.55±2.24 |
| Spleen | 0.085±0.004 | 0.080±0.003 | 2.93±0.25 | 2.90±0.10 |

An unpaired t-test was used for data analysis.

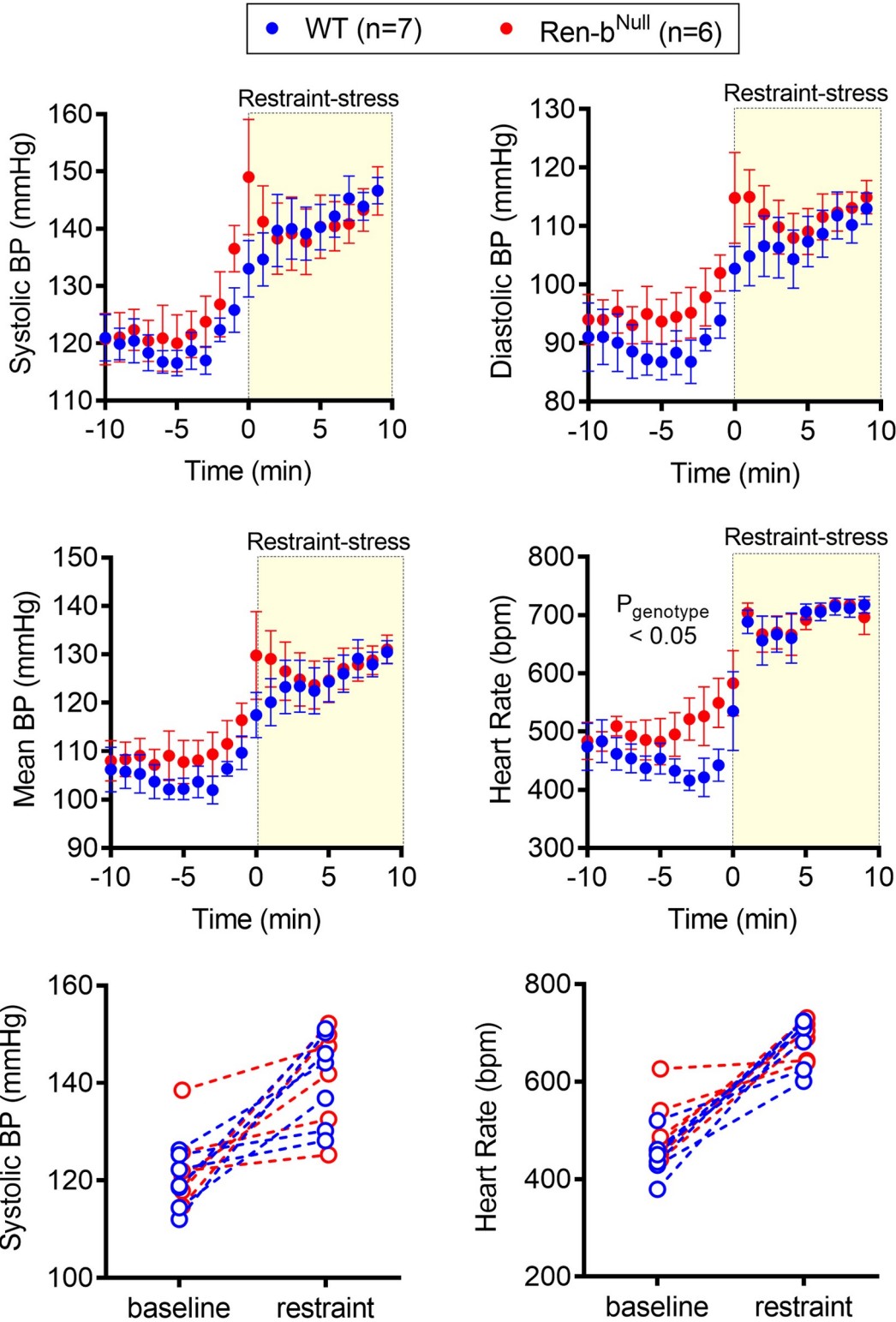

**Fig 5. Blood pressure response to restraint stress (HSD).** Acute responses to restraint-induced blood pressure (BP) and tachycardia in male Ren-b^Null mice and wildtype littermate controls (WT) fed high salt diet. First, telemetric recordings were obtained from high salt diet-fed mice in their home cages at baseline. Following 10 minutes of recording, mice were immediately inserted into an acrylic restrainer and telemetric data were acquired for 10 minutes. One-minute recording was averaged, and the systolic BP, diastolic BP, mean BP, and heart rate (A, B, C, and D, respectively) was expressed as mean

±standard error of the mean. For purposes of clarity and transparency the individual 10 minutes systolic BP and heart rate (E and F, respectively) average of each animal under baseline and restrained conditions was plotted. Data were analyzed by two-way ANOVA with Sidak's multiple comparisons procedure. Adjusted p<0.05 was considered significant.

cohort of male and female mice whether Ren-b[Null] mice exhibit differences in the blood glucose response to restraint-stress. Blood glucose levels were measure in 15 minute-intervals before, during, and after mice were restrained. Control mice were kept in their home cages. Restraint-stress induced a significant increase in blood glucose in both wildtype and Ren-b[Null] mice (P = 0.0002). However, there was no difference between genotypes (Fig 7A and 7B). Plasma renin and glucocorticoids were measured 30 minutes after mice were released from the restraint. As above, unrestrained Ren-b[Null] mice exhibited a trend towards a decrease in plasma renin, but this difference did not reach statistical significance (Fig 7C). No differences in plasma glucocorticoids were observed between animals at the end of the protocol (Fig 7D).

## Expression of renin

In models of hypertension such as deoxycorticosterone-acetate-salt model and previous Ren-b[Null] cohorts exhibiting higher BP, the elevated BP and sympathetic outflow were associated with overexpression of total renin in different nuclei of the brain including the RVLM. Since the current Ren-b[Null] cohorts were normotensive at baseline, we hypothesized that the changes in the phenotype between the previous and the current Ren-b[Null] cohorts are attributable to differences in the expression of renin in the brain. Therefore, a new cohort of mice fed a normal sodium diet was used to quantify expression of total renin in the RVLM and the hypothalamus. Contrary to the elevated total renin level observed in the Ren-b[Null] mice that exhibit elevated blood pressure previously, we found no increase in the renin mRNA levels in the RVLM in the current cohorts (Fig 8A). Interestingly, there was a significant decrease in renin expression in the hypothalamus (Fig 8B). In addition, given that we have previously shown that Ren-b[Null] mice exhibit normal aldosterone levels at baseline but a higher aldosterone secretion when infused with slow-dose Ang II we also measured total renin and renin-a in the adrenal glands. There was no difference in total renin or renin-a among groups (Fig 8C and 8D, respectively) [30].

## Discussion

The main finding of this study is that in a mouse model lacking renin-b, there were significant differences in blood pressure among study cohorts which cannot be attributed to changes in dietary sodium intake nor psychological stress. First, we performed retrospective analyses of the blood pressures in five different Ren-b[Null] cohorts studied over a 3-year period from 2014 to 2017. Interestingly, we observed the cohorts that exhibited the highest blood pressure were housed and manipulated under conditions that might trigger a stress response. Second, since we have previously observed that ablation of renin-b results in elevated renal sympathetic nerve activity we questioned whether mice lacking renin-b would develop impaired salt handling and thus salt sensitivity. Finally, Ren-b[Null] mice on high or low salt diet were further challenged to acute restraint-induced stress. Despite these stimuli, Ren-b[Null] mice did not exhibit a clear difference in blood pressure compared with control littermates.

We previously identified that in models of salt-dependent hypertension such as deoxycorticosterone-salt hypertension there is a downregulation of renin-b expression in the brain with a concomitant elevation of the classical renin isoform (renin-a) encoding preprorenin [32]. Similarly, genetic ablation of renin-b resulted in an augmented expression of renin-a in the RVLM, an important node for autonomic control [22]. We reported that disinhibition of the

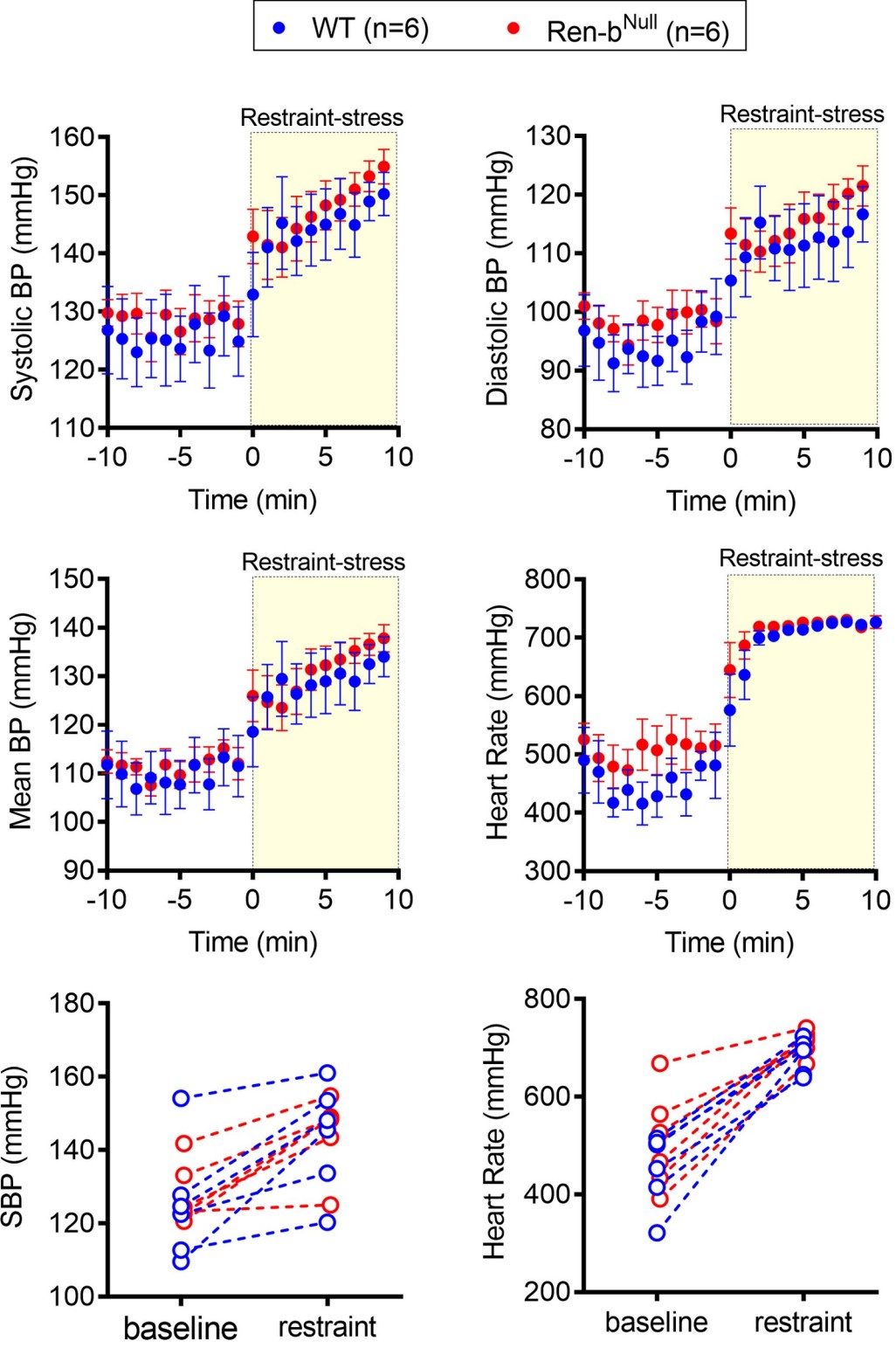

**Fig 6. Blood pressure response to restraint stress (LSD).** Acute responses to restraint-induced blood pressure (BP) and tachycardia in male Ren-b^Null mice and wildtype littermate controls (WT) fed low salt diet. First, telemetric recordings were obtained from low salt diet-fed mice in their home cages at baseline. Following 10 minutes of recording, mice were immediately inserted into an acrylic restrainer and telemetric data were acquired for 10 minutes. One-minute recording was averaged, and the systolic BP, diastolic BP, mean BP, and heart rate (A, B, C, and D, respectively) was expressed as

mean±standard error of the mean. For purposes of clarity and transparency the individual 10 minutes systolic BP and heart rate (E and F, respectively) average of each animal under baseline and restrained conditions was plotted. Data were analyzed by two-way ANOVA with Sidak's multiple comparisons procedure. Adjusted p<0.05 was considered significant.

brain RAS due to renin-b genetic deletion results in a mild elevation of blood pressure, which was accompanied by an elevated renal sympathetic nerve activity. Despite the elevated brain RAS activity, Ren-b[Null] mice exhibited low plasma renin which can be exacerbated with chronic infusion of Ang II [22,30]. However, after successive rounds of backcross breeding, the Ren-b[Null] mice developed a milder phenotype with no apparent blood pressure elevation [30]. Here, we evidenced that these new cohorts of Ren-b[Null] mice lacking elevated blood

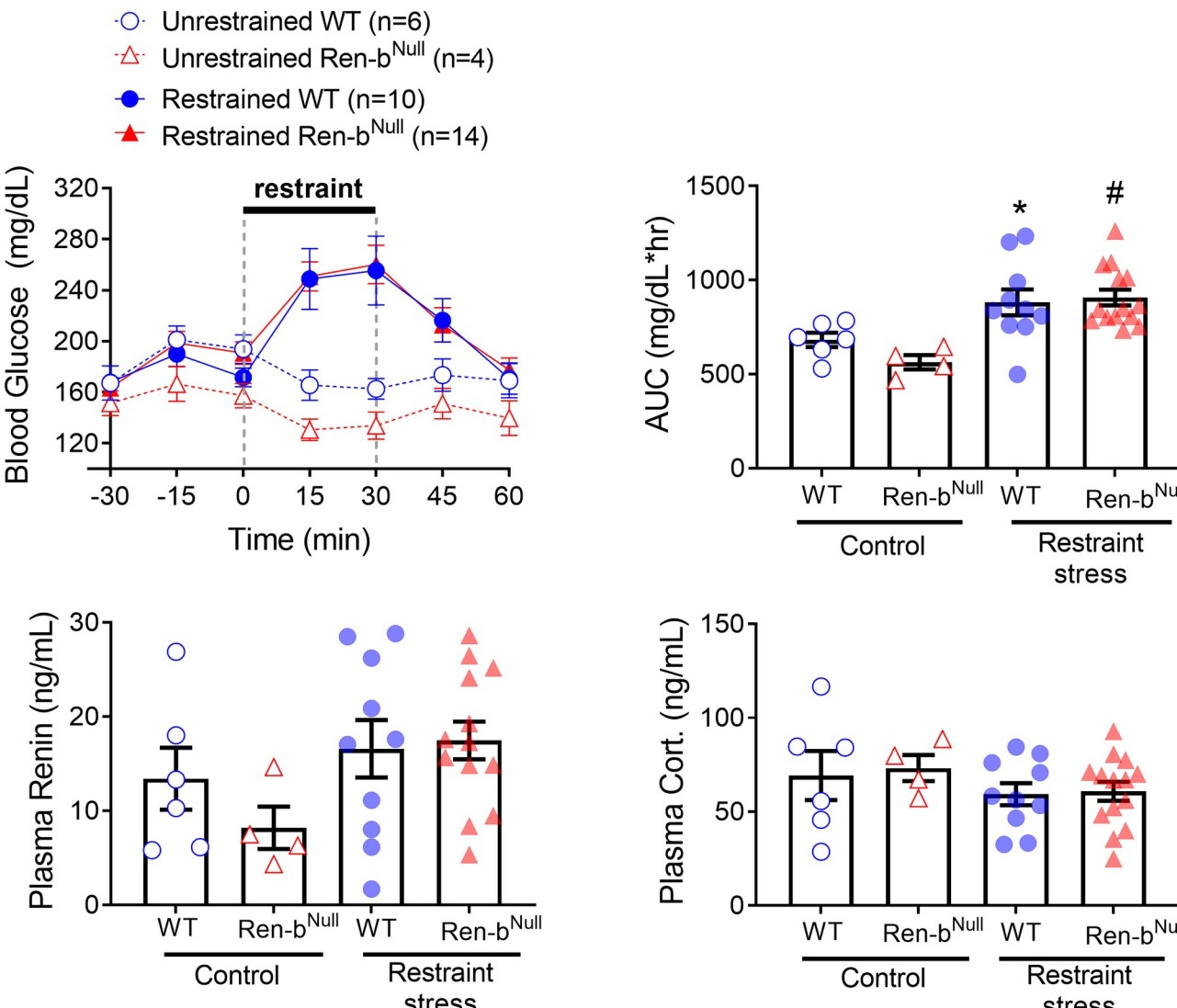

**Fig 7.** A) Blood glucose was determined every 15 minutes before, during, and after restraint-stress in Ren-b[Null] and wildtype littermates (WT) males and females. A group of unrestrained Ren-b[Null] and WT controls, which were kept in their home cages was included. B) The area under the curve (AUC) was calculated and the individual values were plotted. By the end of the protocol the animals were decapitated, and trunk blood was collected for renin (C) and corticosterone (D) determinations. *P<0.05 vs unrestrained WT and #P<0.05 vs unrestrained Ren-b[Null].

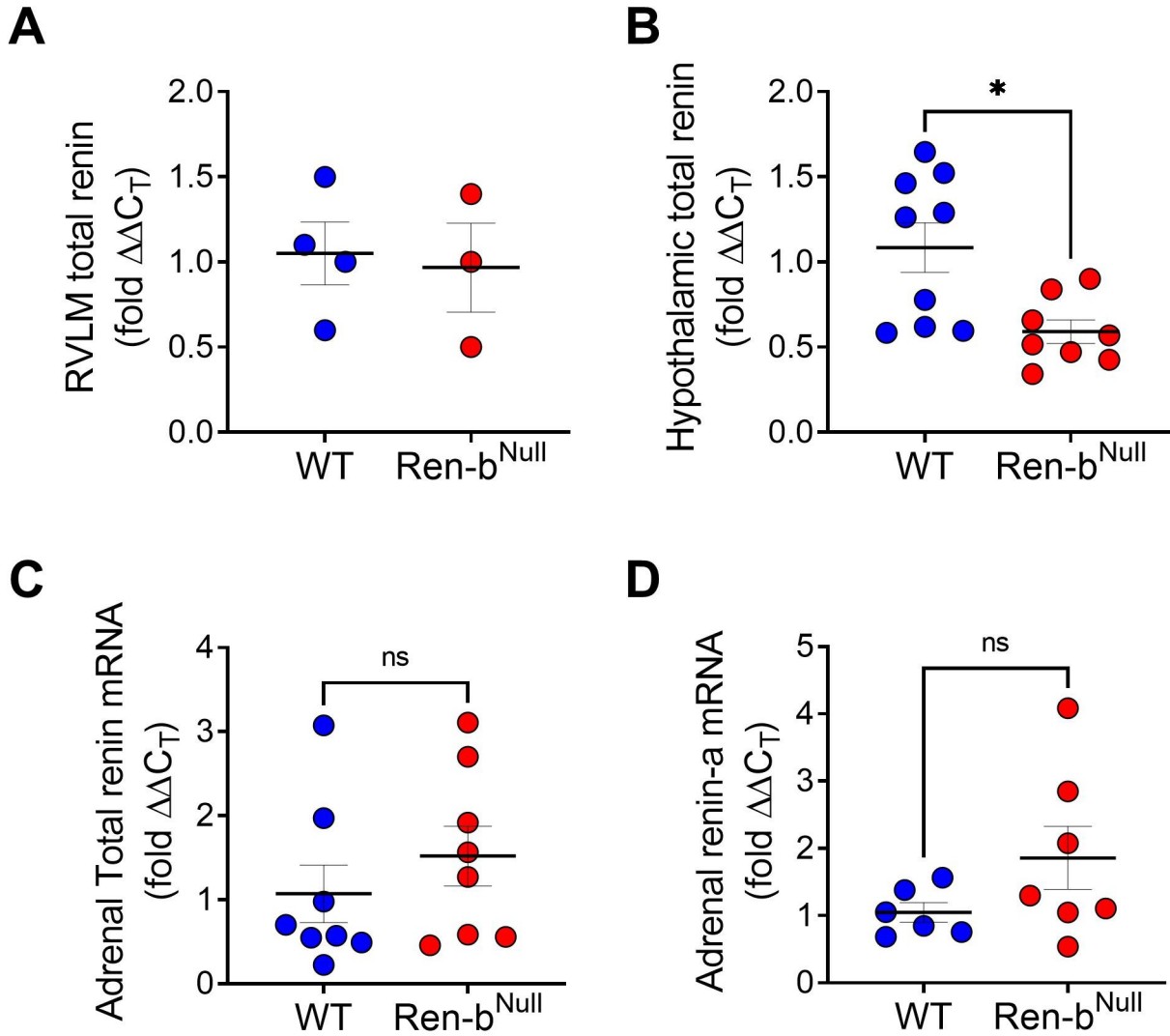

**Fig 8.** Using quantitative reverse transcription PCR total renin expression was measured in A) the rostral ventrolateral medulla (RVLM) and B) the whole hypothalamus. C and D) In the adrenal glands, total renin and renin-a were measured. The wildtype (WT) and Ren-b<sup>Null</sup> mice were fed a normal salt diet. The data were analyzed by unpaired t-test. *P<0.05 vs WT.

pressure exhibit normal total renin expression in the RVLM, but also a paradoxical downregulation of renin in the hypothalamus. Even though this cannot be compared with the previous cohorts–as renin was not measured in the hypothalamus previously–this observation suggests the lack of elevated renin in the RVLM might contribute to the phenotype variability. Furthermore, these alterations in gene expression might be cell specific. We observed that unlike the previous Ren-b<sup>Null</sup> cohorts, there was only a trend toward decrease in plasma renin and no difference in renin expression in the adrenal glands. We considered that environmental or dietary factors might be involved in these phenotypic differences between cohorts.

To test whether ablation of Ren-b results in salt-sensitivity, Renin-b knockout mice and wildtype littermates were first fed a low salt diet for 7 days and subsequently switched to a high salt diet for the next 7 days. A diet high in salt induced a modest increase in blood pressure in both wildtype and renin-b knockout mice. It was recently demonstrated that the C57BL6/J strain is salt-sensitive in response to overactivation of the sympathetic nervous system [33],

but this remains controversial as there are other reports indicating that C57BL6/J are salt resistant [34,35]. Contrary to our prediction, renin-b lacking mice did not exhibit an augmented pressor response to salt. In addition to the data shown in the Fig 3, we also noted that there was no difference in systolic blood pressure when the data were separately analyzed during the light and dark cycle (data not shown). Therefore, we concluded renin-b deficiency is insufficient to induce an increased sensitivity to a pressor response to high salt diet.

Several studies indicate that elevated sodium intake can lead to functional and structural cardiac remodeling independent of changes in blood pressure [12,36]. Here, we observed that Ren-b[Null] mice exhibit a trend towards increased heart weight when fed a high salt diet, but this did not reach statistical significance. This observation is surprising because we showed that Ang-II infusion induced elevated heart weight in Ren-b[Null] mice which was accompanied by induction of pro-inflammatory and pro-fibrotic genes in the myocardium[30]. Studies from Jörg Peters's laboratory have evidenced that cardiac insult triggers renin-b upregulation in the myocardium [37]. Furthermore, myocardial renin-b overexpression results in cardioprotection against ischemic and hypoglycemic conditions [38,39]. Future studies using conditional and tissue-specific models will be required to elucidate the distinct functions of renin-b in different cells including cardiomyocytes.

Given the lack of the phenotype that is observed in the current cohorts of renin-b knockout mice, we hypothesized that ablation of renin-b leads to an enhanced susceptibility to hypertension-eliciting psychological and physical stressors. Under normal conditions, the activity of the brain RAS is relatively low. This is because it is tonically suppressed by renin-b among other factors. However, it has been shown that animals preconditioned to different stressful conditions can lead to brain RAS disinhibition, a condition that can be recapitulated in the Ren-b[Null] model. Importantly, the activation of the brain RAS in response to these environmental factors results in a higher predisposition to hypertension due to enhanced sensitization to a subsequent hypertensive stimulus including Ang-II [40,41]. Therefore, in this current study, we investigated whether renin-b lacking mice are sensitized to an acute restraint-stress induced pressor and metabolic responses. Restraint-stress induced an instantaneous pressor and tachycardiac response in both wildtype and Ren-b[Null] mice. However, there was no significant difference in the magnitude of the response and the timing to maximal blood pressure between them. This suggests that ablation of renin-b does not cause a significant susceptibly to this particular acute stressor.

The main limitation of this study is that the animals were exposed to a single acute exposure to a particular stress induced by immobilization. We observed that restraint induces a dramatic response evidenced by a marked and rapid increase in blood pressure and heart rate. Thus, it is possible that the maximal nature of this response may have caused us to not observe an increased response in Ren-b[Null] mice to a submaximal stress stimulus. It also remains unclear whether a chronic repetitive daily exposure to not only restraint-stress, but also different types of stressors such as noise, wet cage, and vibrations would unmask the cardiovascular and other phenotypes in Ren-b[Null] mice. We recognize that we are limited to recapitulate exactly the environmental conditions that once caused elevated blood pressure in this model, but we are certainly confident that Ren-b[Null] mice are resistant to changes in sodium intake and an acute stressor in our current conditions. This indicates that mechanisms unrelated to salt and acute stress alter the cardiovascular phenotype in mice lacking renin-b. We cannot rule out a possibility that a genetic drift causing a downregulation of Ren-a in the central nervous system occurred in our colony after several breeding cycles. Thus, ongoing studies where Ren-b[Null] mice are backcrossed to a different genetic background will be useful to determine whether the discrepancies in blood pressure occurred due to spontaneous changes in the DNA along intense inbreeding protocols. Finally, other variables such as the microbiome, variations

in temperature and humidity, as well as variations between batches of bedding and chow are difficult to control, but they must be recognized as an important source of variability between experimental cohorts.

## Acknowledgments

The authors gratefully acknowledge the technical assistance of Debbie Davis and Kelsey Wackman with animal care. Transgenic mice were first maintained and genotyped at the University of Iowa Gene Editing Facility supported by grants from the National Institutes of Health and the Carver College of Medicine, and then in the Biomedical Resource Center at the Medical College of Wisconsin.

## Author Contributions

**Conceptualization:** Pablo Nakagawa, Justin L. Grobe, Curt D. Sigmund.

**Data curation:** Pablo Nakagawa, Justin L. Grobe, Curt D. Sigmund.

**Formal analysis:** Pablo Nakagawa, Justin L. Grobe, Curt D. Sigmund.

**Funding acquisition:** Pablo Nakagawa, Curt D. Sigmund.

**Investigation:** Pablo Nakagawa, Javier Gomez, Ko-Ting Lu, Curt D. Sigmund.

**Methodology:** Pablo Nakagawa, Javier Gomez, Ko-Ting Lu, Curt D. Sigmund.

**Project administration:** Pablo Nakagawa, Ko-Ting Lu, Curt D. Sigmund.

**Resources:** Pablo Nakagawa, Justin L. Grobe, Curt D. Sigmund.

**Software:** Pablo Nakagawa, Curt D. Sigmund.

**Supervision:** Pablo Nakagawa, Justin L. Grobe, Curt D. Sigmund.

**Validation:** Pablo Nakagawa, Justin L. Grobe, Curt D. Sigmund.

**Visualization:** Pablo Nakagawa, Justin L. Grobe, Curt D. Sigmund.

**Writing – original draft:** Pablo Nakagawa.

**Writing – review & editing:** Pablo Nakagawa, Javier Gomez, Ko-Ting Lu, Justin L. Grobe, Curt D. Sigmund.

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
