## [Decision Letter · Decision Letter 0]

26 Apr 2021

PONE-D-21-11323

Studies of Salt and Stress Sensitivity on Arterial Pressure in Renin-b Deficient Mice

PLOS ONE

Dear Dr. Nakagawa,

Thank you for submitting your manuscript to PLOS ONE. After careful consideration, we feel that it has merit but does not fully meet PLOS ONE’s publication criteria as it currently stands. Therefore, we invite you to submit a revised version of the manuscript that addresses the points raised during the review process.

We look forward to receiving your revised manuscript.

Kind regards,

Michael Bader

Academic Editor

PLOS ONE

Journal Requirements:

'I have read the journal's policy and the authors of this manuscript have the following competing interests:

C.D. Sigmund is a member of the Scientific Advisory Board for Ionis Pharmaceuticals.

His contributions to that board are unrelated to the content of this article.

The other authors report no conflicts.'

a. Please confirm that this does not alter your adherence to all PLOS ONE policies on sharing data and materials, by including the following statement: "This does not alter our adherence to  PLOS ONE policies on sharing data and materials.” (as detailed online in our guide for authors http://journals.plos.org/plosone/s/competing-interests).  If there are restrictions on sharing of data and/or materials, please state these.

Please note that we cannot proceed with consideration of your article until this information has been declared.

Reviewers' comments:

Reviewer's Responses to Questions

**Comments to the Author**

1. Is the manuscript technically sound, and do the data support the conclusions?

Reviewer #1: No

Reviewer #2: Partly

Reviewer #3: Yes

2. Has the statistical analysis been performed appropriately and rigorously? 

Reviewer #1: No

Reviewer #2: Yes

Reviewer #3: Yes

3. Have the authors made all data underlying the findings in their manuscript fully available?

Reviewer #1: No

Reviewer #2: Yes

Reviewer #3: Yes

4. Is the manuscript presented in an intelligible fashion and written in standard English?

Reviewer #1: Yes

Reviewer #2: Yes

Reviewer #3: Yes

5. Review Comments to the Author

Reviewer #1: Nakagawa et al. come up with a very peculiar concept, i.e., that renin-b suppresses renin-a. What would be the logic of having such a system, and how exactly does this happen? The body has many ways to up- or downregulate renin, including BP, salt and Ang II. Since renin-b remains in the cell, how does it affect renin-a? And then apparently this only applies to cells which generate both renin-a and renin-b? Do we have evidence that both occur in the same cell.

The authors hypothesize that mice lacking renin-b would be more responsive to an acute environmental stressor as consequence of brain RAS disinhibition. In the absence of renin-b, the ‘’disinhibition’’ must already be absent, i.e., renin-a is permanently upregulated? Do they have evidence for this in their various cohorts, i.e., might the different response relate to different upregulations of renin-a? And if so, why do we only see the consequence of this during acute stress? This implies that without stress the upregulated renin-a is still suppressed: by what? What then exactly happens during acute stress (mechanistically)? I fail to understand this.

What about the consequence of renin-b disappearance in the adrenal in the renin-b Null mice? Does it upregulate renin-a at that site? Please provide such data in your model. The renin-b Null mice display proteinuria: why? This occurs in the absence of hypertension. Ren-b Null mice also exhibited an elevated heart rate at baseline, although this is very hard to see in Figure 5. What kind of test was used to reach this conclusion, particularly if it applies only to a few data points?

The authors refer to previous data on renin ‘’showing a trend to decrease’’ after high-salt. I am not sure what this means. At P=0.094 this cannot be stated (nor is it confirmed here), nor does the ‘’trend’’ toward increased heart weight in Table 2. Please refrain from drawing conclusions if P>0.05, and only discuss/show novel data. Figure 7 also shows no difference between WT and renin-b Null mice.

Taken together, after Figure 1, it is very hard to see what exactly is the difference between the WT and renin-b Null mice. It seems there is no difference after all? The authors refer to a ‘’decrease’’ in plasma renin, which is not apparent/significant however. There are no differences in salt sensitivity, and even the stress response does not seem to be different. They mention a trend towards increase in heart weight, but this too is NS. The authors suggest that the phenotype is ‘’milder’’ (in reality it is absent?), and speculate about genetic drift. After reading and re-reading this paper, I fail to see what the message is. It seems that the phenotype is entirely lacking. Preferably, the authors first provide a plausible mechanism by which all of this might occur: how does renin-b affect renin-a.

Reviewer #2: The study Nakagawa et al. starts discussing a controversial blood pressure phenotype found in their previous studies using mice lacking the intracellular renin isoform (Ren-b). The authors suspected the phenotype discrepancy could potentially be related to stressful experimental conditions in previous experiments that would have induced high blood pressure in Ren-b null mice. To test this hypothesis, the impact of low and high salt diets on blood pressure in mice lacking Ren-b was investigated. Additionally, the authors submitted these mice while on low and high salt diets to an acute restraint-stress protocol to evaluate cardiovascular changes. Overall, this study could not show significant cardiovascular changes among Ren-b deleted animals in comparison to controls, but it uses an interesting experimental approach to test cardiovascular physiology in Ren-b null mice. Below are some comments that should help to improve the work.

Major

1) At the end of the abstract the following sentence is written “These studies highlight a complex mechanism that masks/unmasks roles for renin-b in cardiovascular physiology”. What is meant by this sentence, because there was no apparent cardiovascular modulatory effect of Ren-b deletion on high salt diet and stress, neither was proven that stress was correlated to the blood pressure discrepancy in previous studies

2) The authors performed retrospective analyses of the light cycle systolic blood pressure, and argued the blood pressure differences may be related to the room dimensions and people traffic that were not favorable during the recordings of cohorts 1-4. Looking at the original publication it is evident that only Sys pressure is increased only during day time in Ren-b null mice. This information should be included in the present study at least in the text. If light cycle Sys pressure was increased, why this parameter was not investigated separately in the present study rather than the daily average. This should be at least mentioned if there is no difference at baseline as well as during the salt diet challenge. Additionally, do the authors have data recorded during the weekend or holidays, when the room was likely empty, as it seems to be in the dark period? Besides, what is the author’s opinion about the final blood pressure phenotype of Ren-b null mice. Are the data from the previous studies invalid?

3) The authors mentioned that DOCA-salt reduces central Ren-b and increases Ren-a expression, why they did not investigate the mRNA levels in WT brains after high salt diet.

4) The authors wrote the following sentence in the abstract. “When renin-b deficient mice were exposed to a high salt diet for a longer duration (4 weeks), there was a trend for increased myocardial enlargement in Ren-b null mice when compared with control mice”. Indeed, it looks like there was a trend to increased heart weight. However, an age matched group on a normal salt diet should be included to show the initial cardiac weight, to check if this trend was not independent of salt.

5) Why the authors decided to use different time points upon high salt-diet to evaluate BP, stress and morphology? Blood pressure was investigated 1 week after starting the high salt diet, stress 2 weeks and the morphology 4 weeks after.

6) Please give a rational reason why not to investigate the blood pressure response to stress using mice on a normal salt diet too, instead of under low and high salt diets, in particular, because the authors suspected stress could have induced elevation in BP in the cohorts 1-4 in which the animals where likely on a normal salt diet. In the discussion it seems like the experiments were performed in animals on a normal salt diet but this is clearly not the case.

7) In the results it says “Interestingly, an early cohort of Ren-b null mice exhibited higher urinary protein excretion compared with wildtype mice at baseline. The level of protein excretion was maintained in Ren-b null mice after high salt diet. On the contrary, high salt diet increased urinary protein excretion in WT mice.” What is meant by an early cohort of animals? Were those mice hypertensive and, therefore, not from this study?

8) In their discussion the authors mentioned “Similarly, genetic ablation of renin-b resulted in an augmented expression of renin-a in several brain nuclei including the rostral ventrolateral medulla [20]. In the original publication cited, only the RVLM presented significantly increased expression of Ren-a, the other nuclei had a tendency to have increased mRNA. Also, overall brain mRNA level of Ren-a in Ren-b null mice seems not be to altered. Please be more precise when citing data from previous studies in particular those from the own group.

9) Plasma renin activity values shown in figure 7, baseline or after stress, are not higher than 20 ng/mL in average in both WT and Ren-bnull, while animals submitted to a high salt diet in Table 1 WT present 44.8 and Ren-bNull 24.3. However, high salt diet should suppress renin release and peripheral Ang II production, please discuss this surprising result.

10) Plasma renin activity values shown in figure 7, baseline or after stress, are not higher than 20 ng/mL in average in both WT and Ren-bnull, while animals submitted to a high salt diet in Table 1 WT present 44.8 and Ren-bNull 24.3. However, high salt diet should suppress renin release and peripheral Ang II production, please discuss this surprising result.

Minor

1) All Graphs should be marked with A, B, C… on top, otherwise it is confusing to find the graph referred in the text and/or figure legend. Please standardize it, currently some figure legend refer to A,B,C… while others not.

2) Figure 3, top panel: blue dot legend and n are missing.

3) Figure 7 legend has no general description of content as the previous one.

4) On page 3 references should be added to the following sentence “Numerous studies have identified neurogenic actions of elevated salt intake”

5) This sentence in page 3 seems to be incomplete: “For instance, elevated dietary sodium intake sensitizes sympathetic neurons located in the in response to a stimulus [9]”.

6) In the methodology it is written “A cohort of Ren-b Null mice without telemetry implants was placed in metabolic cages before and after 3 weeks on a high salt diet”, and table 1 legend says “Table 1: Male Wildtype (WT) and Ren-b null mice were housed in metabolic cages to measure food intake, feces weight, water intake, urine volume, and urinary protein excretion at baseline and after 4 weeks on high salt diet (HSD, 4% NaCl)”. What is correct, 3 or 4 weeks after HSD?

7) Table 2 – The legend says “Body weight (left column) and tissue weights normalized by body weight (right column)”. The body weight is not shown in this table. The left column shows tissue organ weights, correct?

8) Methodology of Experiment 2 – please provide more information regarding this experiment, particularly the exact tonicity of the injected solutions, the time course of the experiment and how the urine was sampled.

Reviewer #3: The authors compared the salt- and stress- sensitivity of renin-b null and wild type mice with respect to blood pressure, heart rate and renal and metabolic parameters. Their hypothesis was, that under salt- or restrain-stress renin-b null mice may have elevated blood pressure and/or heart rates.

The rational to perform the present study was as follows: The authors previous showed that the genetic removal of renin-b triggered a mild increase in blood pressure and sympathetic nerve activity. The authors previously hypothesised that renin-b maintains brain RAS activity at normal physiological levels by suppressing the classical renin isoform and thus, renin-b deletion would des-inhibit a brain RAS. However, in later studies with other cohorts of renin-b null and wild type mice differences in blood pressure or heart rate were not observed any more. The authors hoped to attribute these discrepancies to differences responses to salt intake and/or stress.

The experiments were very well designed and performed. From the methodological point of view, I have only minor criticism (see below).

However, with respect to the original hypothesis (renin-b inhibits renin-a expression in the brain and des-inhibition of renin-a in the absence of renin-b increases blood pressure) it would be essential to measure renin-a expression again in the brain (or in specific brain areas), i. e. to demonstrate that renin- a mRNA it is still elevated (or not). Only thus the original hypothesis can be supported or must be discarded.

Minor comments:

I would not rely on „trends”( heart weight/BW; plasma renin levels) – moreover, a p-value around 0,09 may not even be considered as a trend.

Basal urinary protein excretion were significantly higher in renin-b null – with no further increase under HSD. Please explain possible reasons.

Table 2: please add p-values

Heart rate Fig. 5: the basal heart rates were initially not different (minutes -10 to -5). This was followed by a transient increase, but just before restrain-stress there was again no difference when compared to wild type.

6. PLOS authors have the option to publish the peer review history of their article (what does this mean?). If published, this will include your full peer review and any attached files.

Reviewer #1: No

Reviewer #2: No

Reviewer #3: **Yes: **Joerg Peters

---

## [Author Response · Author response to Decision Letter 0]

21 Jun 2021

Please, see our responses to the reviewers and the editor in the attached document.

---

## [Decision Letter · Decision Letter 1]

7 Jul 2021

PONE-D-21-11323R1

Studies of Salt and Stress Sensitivity on Arterial Pressure in Renin-b Deficient Mice

PLOS ONE

Dear Dr. Nakagawa,

Thank you for submitting your manuscript to PLOS ONE. After careful consideration, we feel that it has merit but does not fully meet PLOS ONE’s publication criteria as it currently stands. Therefore, we invite you to submit a revised version of the manuscript that addresses the points still raised by all three reviewers.

We look forward to receiving your revised manuscript.

Kind regards,

Michael Bader

Academic Editor

PLOS ONE

Journal Requirements:

Reviewers' comments:

Reviewer's Responses to Questions

**Comments to the Author**

1. If the authors have adequately addressed your comments raised in a previous round of review and you feel that this manuscript is now acceptable for publication, you may indicate that here to bypass the “Comments to the Author” section, enter your conflict of interest statement in the “Confidential to Editor” section, and submit your "Accept" recommendation.

Reviewer #1: (No Response)

Reviewer #2: All comments have been addressed

Reviewer #3: (No Response)

2. Is the manuscript technically sound, and do the data support the conclusions?

Reviewer #1: Yes

Reviewer #2: Yes

Reviewer #3: Yes

3. Has the statistical analysis been performed appropriately and rigorously? 

Reviewer #1: Yes

Reviewer #2: Yes

Reviewer #3: Yes

4. Have the authors made all data underlying the findings in their manuscript fully available?

Reviewer #1: No

Reviewer #2: Yes

Reviewer #3: (No Response)

5. Is the manuscript presented in an intelligible fashion and written in standard English?

Reviewer #1: Yes

Reviewer #2: Yes

Reviewer #3: Yes

6. Review Comments to the Author

Reviewer #1: All questions have been addressed, and the authors provided novel data on renin a expression in the adrenal. Please add these to the MS, and discuss them accordingly. These are highly relevant, as their concept is that renin b somehow suppresses renin a. Apparently, this is not true in the adrenal? Why would this only happen in the brain?

Reviewer #2: (No Response)

Reviewer #3: My comments have been addressed. The lack of change in renin (renin-a) expression fits to the lack of change in blood pressure. However, there was a decrease - not increase of renin (a) in the hypothalamus. This contradicts the authors conclusion that renin-b decreases renin-a expression. Furthermore, if renin-a in the hypothalamus has any role at all, this decrease should have some consequences. This need to be stated and discussed.

7. PLOS authors have the option to publish the peer review history of their article (what does this mean?). If published, this will include your full peer review and any attached files.

Reviewer #1: No

Reviewer #2: No

Reviewer #3: No

---

## [Author Response · Author response to Decision Letter 1]

12 Jul 2021

We carefully addressed the reviewers' inquiries. Our responses are attached in a separate word document.

---

## [Editor Report · Decision Letter 2]

15 Jul 2021

Studies of Salt and Stress Sensitivity on Arterial Pressure in Renin-b Deficient Mice

PONE-D-21-11323R2

Dear Dr. Nakagawa,

We’re pleased to inform you that your manuscript has been judged scientifically suitable for publication and will be formally accepted for publication once it meets all outstanding technical requirements.

Kind regards,

Michael Bader

Academic Editor

PLOS ONE
---

## [Editor Report · Acceptance letter]

19 Jul 2021

PONE-D-21-11323R2 

Studies of Salt and Stress Sensitivity on Arterial Pressure in Renin-b Deficient Mice 

Dear Dr. Nakagawa:

I'm pleased to inform you that your manuscript has been deemed suitable for publication in PLOS ONE. Congratulations! Your manuscript is now with our production department. 

Kind regards, 

on behalf of

Prof. Michael Bader 

Academic Editor

PLOS ONE